# Thermal Shift Assay for Small GTPase Stability Screening: Evaluation and Suitability

**DOI:** 10.3390/ijms23137095

**Published:** 2022-06-26

**Authors:** Kari Kopra, Salla Valtonen, Randa Mahran, Jonas N. Kapp, Nazia Hassan, William Gillette, Bryce Dennis, Lianbo Li, Kenneth D. Westover, Andreas Plückthun, Harri Härmä

**Affiliations:** 1Department of Chemistry, University of Turku, Henrikinkatu 2, 20500 Turku, Finland; sallamaria.valtonen@gmail.com (S.V.); randa.r.mahran@utu.fi (R.M.); nazia.n.hassan@utu.fi (N.H.); harri.harma@utu.fi (H.H.); 2Department of Biochemistry, University of Zurich, Winterthurerstrasse 190, 8057 Zurich, Switzerland; j.kapp@bioc.uzh.ch (J.N.K.); plueckthun@bioc.uzh.ch (A.P.); 3Leidos Biomedical Research, Inc., Frederick National Laboratory for Cancer Research, 8560 Progress Dr., Frederick, MD 21702, USA; gillettew@mail.nih.gov; 4Departments of Biochemistry and Radiation Oncology, University of Texas Southwestern Medical Center at Dallas, 5323 Harry Hines Blvd, L4.270, Dallas, TX 75390, USA; brycedennis@gmail.com (B.D.); lianbo.li@utsouthwestern.edu (L.L.); kenneth.westover@utsouthwestern.edu (K.D.W.)

**Keywords:** differential scanning fluorimetry (DSF), thermal stability assay (TSA), GTPase, KRAS, Protein-Probe, SYPRO Orange

## Abstract

Thermal unfolding methods are commonly used as a predictive technique by tracking the protein’s physical properties. Inherent protein thermal stability and unfolding profiles of biotherapeutics can help to screen or study potential drugs and to find stabilizing or destabilizing conditions. Differential scanning calorimetry (DSC) is a ‘Gold Standard’ for thermal stability assays (TSA), but there are also a multitude of other methodologies, such as differential scanning fluorimetry (DSF). The use of an external probe increases the assay throughput, making it more suitable for screening studies, but the current methodologies suffer from relatively low sensitivity. While DSF is an effective tool for screening, interpretation and comparison of the results is often complicated. To overcome these challenges, we compared three thermal stability probes in small GTPase stability studies: SYPRO Orange, 8-anilino-1-naphthalenesulfonic acid (ANS), and the Protein-Probe. We studied mainly KRAS, as a proof of principle to obtain biochemical knowledge through TSA profiles. We showed that the Protein-Probe can work at lower concentration than the other dyes, and its sensitivity enables effective studies with non-covalent and covalent drugs at the nanomolar level. Using examples, we describe the parameters, which must be taken into account when characterizing the effect of drug candidates, of both small molecules and Designed Ankyrin Repeat Proteins.

## 1. Introduction

In recent years, thermal shift assays (TSAs) have become a popular tool, especially for early drug screening and development, but also for determining optimal buffer compositions [1,2,3]. Differential scanning calorimetry (DSC), the TSA ‘Gold Standard’, has limited potential for these higher throughput purposes, and thus other biophysical techniques, such as differential scanning fluorimetry (DSF), have become increasingly practical and popular options [3,4,5]. DSF offers a cost-effective alternative, as it can be performed in parallel; DSC experiments are typically run sequentially. DSC is considered to be the most precise method for protein unfolding measurements and can in some cases lead to the direct measurement of thermodynamic parameters (ΔH, TΔS, ΔG) in addition to melting temperature (T_m_). These factors are outweighed by the benefit afforded by a higher throughput method such as DSF [4,6,7]. Another consideration is that T_m_ values may vary between approaches [3,4].

Within the DSF technique, scan rate and dye selection affect the accuracy and values gathered. There is limited knowledge regarding the exact mechanisms of how each dye interacts with the target protein, and thus the suitability of each DSF dye also varies from case to case [2,4,8,9,10]. There are several types of external fluorescent DSF dyes that function through different mechanisms, and they are often used for different types of applications. Rotational dyes sense the viscosity changes in their vicinity and produce higher signals when the environment becomes more viscous. [2,11]. Thus, these dyes, e.g., Proteostat^®^ and thioflavin T, are most often used to monitor aggregation but also surfactant and other buffer composition-related aspects [2,11,12]. On the other hand, environmental polarity sensing dyes, e.g., SYPRO Orange and 8-anilino-1-naphthalenesulfonic acid (ANS), are useful for protein thermal stability and interaction monitoring. These dyes are quenched in an aqueous environment, but upon protein denaturation, where the inner hydrophobic amino acids are exposed, the non-polar binding environment intensifies the fluorescence of these dyes [2,4,12]. Unlike rotational dyes, which are used for fibrillation and aggregation monitoring, polarity-sensing dyes have broader applicability [2,4,13]. These dyes can detect aggregation but are mainly used for protein thermal stability monitoring. SYPRO Orange, in particular, has gained popularity, as it is directly compatible with most common qPCR equipment [1,2,4].

While the robustness of DSF enables broad applicability in drug discovery, there are some limitations related to this technique. One comes from the non-specific nature of the extrinsic DSF dyes, which mainly utilize weak non-covalent interactions for binding to their target [1,2,3,4,8]. However, by monitoring the intrinsic tryptophan fluorescence at 330 nm and 350 nm, the use of extrinsic dyes can be overcome. This technique enables not only the detection of protein unfolding, but also the refolding, which is not usually measurable using conventional DSF [2,14,15]. In addition, denaturation monitoring from red-shifted tryptophan fluorescence is more compatible with detergents compared to extrinsic dyes like SYPRO Orange [2,16,17]. However, tryptophan is a relatively rare amino acid and can vary in number and position from protein to protein. In addition, fluorescence at low wavelengths (250–350 nm), as with tryptophan or ANS, is subject to background fluorescence noise related to assay materials or experimental compounds [18]. Thus, the dye selection and knowledge about assayed materials has become even more important. Careful attention to these details might alleviate problems related to the thermal profile and reduce the occurrence of incorrect conclusions by lowering background fluorescence.

We previously developed a novel DSF-like method called the Protein-Probe [19,20,21,22]. The method is based on a Eu^3+^-labeled peptide probe, where a negatively charged peptide has minimal interaction with a folded protein, and the time-resolved luminescence (TRL) signal is low in the modulation solution containing a cyanine dye as quencher. The TRL signal increases upon unfolding of the protein because of an increased interaction of the hydrophobic core with the peptide and greater distance of the Eu^3+^-label from the quencher. This technique can avoid many of the problems related to conventional DSF dyes. The Protein-Probe enables similar stability, interaction, and buffer solution composition studies as conventional external thermal dyes, but with significantly improved sensitivity [19,20,21,22]. In addition, we have extensively studied KRAS and other GTPase proteins to enable monitoring of their functional properties [23,24,25,26]. KRAS has gained high interest as a drug target, and multiple inhibitors applicable as tool compounds have been developed. Due to the enzymatic nature of KRAS, it is a structurally flexible protein [27,28,29]. Activating mutations in all RAS proteins are clustered around the nucleotide binding pocket, at amino acids 12, 13, 61, 117, and 146. The hotspot mutation positions of G12 and G13 are located on the P-loop, have a nucleotide-stabilizing role, and in the case of G13, a significant effect on RAS activity [25,30]. On the other hand, Q61, located at the switch II region, participates in the conformational changes crucial for RAS inactivation, having the lowest hydrolysis rates among all KRAS alleles [25,30]. Lately, KRAS G12C has been in the spotlight and KRAS G12C-targeted molecules have been developed as potential cancer treatments [31,32,33].

Thus we selected the KRAS of other GTPase proteins as our models to address how point mutations, protein and nucleotide concentration, buffer components, and inhibitors affect target protein stability. Here, we show that the improved sensitivity of the Protein-Probe method allows more differentiating studies of KRAS mutants, in comparison to existing TSA dyes. We also highlight the potential problems and shortcomings of DSF approaches, which emphasize the need for careful assay optimization and the significance of small details in data collection, interpretation, and comparison.

## 2. Results and Discussion

### 2.1. Thermal Stability Is Buffer and Target GTPase Concentration Dependent

DSF is a widely used technique due to its simplicity. However, results obtained in various studies are rarely comparable, even when the assays use the same DSF probe, usually SYPRO Orange. Previously, we demonstrated a new and highly sensitive method for TSA, the Protein-Probe technique. The Protein-Probe senses the protein stability and factors affecting its stability, but with significantly improved sensitivity compared to dyes like SYPRO Orange [19,20]. Partly due to improved sensitivity, we found it nearly impossible to evaluate and compare it with previously published results. Nevertheless, by using KRAS and other GTPases as model systems, we were able to address critical factors that were not clearly addressed in previous publications. Mutation positions of the studied KRAS constructs are marked in Appendix A.

It is widely known that buffer composition drastically affects protein stability. This is especially true with proteins like KRAS, in which Mg^2+^ maintains the protein in its stable nucleotide-bound form. Thus, we first studied the KRAS WT thermal stability in the presence or absence of MgCl_2_ and the effect of EDTA as a chelating agent (Figure 1A, Appendix A). With 1 mM Mg^2+^, KRAS WT showed the expected stability (T_m_ = 60.1 ± 0.3 °C) within the broad range of previously reported values (53.3–74.4 °C) [34,35,36,37]. Next, we repeated the assay with 10 mM MgCl_2_ and observed a further KRAS-stabilizing effect (T_m_ = 64.7 ± 0.5 °C). EDTA addition resulted in no change to observed T_m_ values with KRAS WT in comparison to the assay performed without MgCl_2_. In both cases, T_m_ values of 48.9 ± 0.4 °C and 49.2 ± 0.4 °C were significantly lower in comparison to those detected with MgCl_2_ (Figure 1A).

Next, we tested if the same was true for the KRAS mutants G13D and Q61R and if assays could be performed with or without 1 mM MgCl_2_, referring to nucleotide-loaded and nucleotide-free KRAS, respectively (Figure 1B, Appendix A). These mutants were selected, as G13D is known to have an exceptionally fast intrinsic nucleotide exchange rate, while the intrinsic exchange of Q61R is very low [25,27,30,38]. The observed thermal stability inversely correlated with the rate of intrinsic nucleotide exchange, as the fast exchange G13D (T_m_ = 47.7 ± 0.4 °C) showed lower stability, and the slow exchange Q61R (T_m_ = 66.5 ± 0.5 °C) displayed higher stability compared to the KRAS WT in the same conditions (Figure 1B). While we observed similar T_m_ for G13D without Mg^2+^ and with EDTA (T_m_ = 43.4 ± 0.6 °C and 43.6 ± 0.8 °C), we observed a pronounced difference between Mg^2+^ and EDTA for Q61R (T_m_ = 59.2 ± 0.7 °C and 50.6 ± 1.0 °C). Clearly, Q61R needed an additional 1 mM EDTA to reach the partially nucleotide-free state. These results indicate a clear link between KRAS thermal stability and the intrinsic nucleotide exchange related to Mg^2+^ binding.

We tested this hypothesis with other KRAS hotspot mutants, G12D, G12C, and Q61L, in the presence of 1 mM MgCl_2_ (Appendix A). All of these displayed similar thermal stability as KRAS WT and thus were between the stability of G13D and Q61R (Figure 1C, Appendix A). These mutants are reported to have a similar rate of intrinsic nucleotide exchange activity as KRAS WT; this suggests that the intrinsic nucleotide exchange activity linked to Mg^2+^ coordination might be the main factor determining thermal stability [27]. Thus, the Mg^2+^ concentration, even when present in great excess, used in each assay might cause significant variations in T_m_ values reported. 

The assays of KRAS mutants were performed at a lower protein concentration than that used in the MgCl_2_ assays (Figure 1A), and we saw a slight shift in T_m_ values when these two assays were compared. When the results obtained with the Protein-Probe were further compared to those with SYPRO Orange, we saw consistently slightly higher T_m_ values using the Protein-Probe. As in these assays, the KRAS concentration was the main determinant for these changes; we titrated KRAS G12C using the Protein-Probe (0.01–1.25 µM) and SYPRO Orange (1–30 µM). The concentration-related shift in T_m_ was observed with SYPRO Orange (55.7–60.3 °C) but especially with the Protein-Probe (Figure 1D, Appendix A). When the KRAS G12C concentration was increased 25-fold, from 50 to 1250 nM, T_m_ values shifted from 61.8 ± 0.5 °C to 53.4 ± 0.6 °C, respectively. We confirmed this result using the Protein-Probe with the Q61R and Q61L mutants, and in these cases the thermal stability decreased by 11.3 °C and 11.8 °C upon KRAS concentration increase from 50 to 1250 nM, respectively (data not shown). These results indicate that care should be taken when comparing T_m_ values from different studies.

When testing various RAS constructs, we found that the sensitivity of the Protein-Probe is related to the exact type of KRAS construct. Even though KRAS is a small protein, it is often studied using only the so-called G-domain, a truncated form that does not contain the *C*-terminal hypervariable region of residues 170–188. Two G-domain constructs were assessed here using Ac-KRAS and iMet-KRAS [39]. In Ac-KRAS, the *N*-terminal initiator methionine (iMet) was cleaved off, and threonine was *N*-acetylated. On the other hand, iMet-KRAS contained an additional non-native glycine as the second amino acid in its *N*-terminus. With the Protein-Probe, 50-fold (Ac-KRAS) and 10-fold (iMet-KRAS) higher concentrations were needed for comparable signal strength to full-length KRAS (Appendix A). However, not only the length but also *N*-terminal processing affected the detectability, as there is a clear difference between Ac-KRAS and iMet-KRAS (Appendix A). This indicates the increase in flexibility of iMet-KRAS in comparison to Ac-KRAS, even though there is no clear difference in T_m_ values, suggesting no changes in nucleotide and Mg^2+^ coordination [39]. In addition, the assays performed with the full-length HRAS and NRAS indicate that the size of the construct is not the main determinant of protein detectability using the Protein-Probe, as both of these constructs were less visible compared to full-length KRAS. Interestingly, the T_m_ values were all higher in comparison to KRAS (54.4 ± 0.6 °C) when assayed at the same conditions using 1 µM KRAS concentration. Detected values for iMet-KRAS, Ac-KRAS, HRAS, and NRAS were 66.9 ± 0.2, 66.0 ± 1.0, 59.0 ± 0.3, and 64.0 ± 0.1 °C, respectively (Appendix A).

In conclusion, not only the length of the RAS construct but also mutations and flexibility might affect protein detectability, especially at nM concentrations. SYPRO Orange could not be used at nM KRAS concentrations, but its optimal signal-to-background (S/B) ratio was found at the very high KRAS concentration of 10 µM (data not shown). In addition, the second dye tested, 8-anilinonaphthalene-1-sulfonic acid (ANS), requires KRAS concentration levels of 2–10 µM. With µM KRAS used in SYPRO Orange and ANS assays, the relatively high concentration seems to equalize differences, which were visible using the Protein-Probe assays at nM concentrations.

### 2.2. Increased Nucleotide Concentration Increases GTPase Stability

Many of the commercially available GTPases are stored in buffers with an excess of GDP to increase their stability. In addition, an excess of GMP (guanosine-5’-monophosphate) is often used to thermally and enzymatically stabilize the nucleotide-free form of RAS, as its effect on biochemical RAS activity assays can be considered negligible due to the low binding affinity of GMP [40]. Results obtained with or without Mg^2+^ indicate that the stabilizing effect is mediated through intrinsic nucleotide exchange or through Mg^2+^-independent nucleotide binding, occurring with lowered affinity [41,42,43]. These processes have been reported to be linked but to occur with different rates, as Mg^2+^ is in very rapid equilibrium with the solvent. Processes are also found to vary between different GTPases and their mutants, and the degree to which stabilization occurs should vary as well [41,42,43].

To test this, we first performed a GDP (guanosine-5’-diphosphate) titration (0–0.9 mM) using KRAS G13D, a highly unstable mutant due to A59 placement in the Mg^2+^ binding site (Figure 2A) [27,44]. As anticipated, a significant increase in KRAS G13D stability was observed with the Protein-Probe as a response to increased GDP concentration, saturating at approximately 300 µM GDP. We confirmed this stability increase with SYPRO Orange using the minimally possible KRAS concentration for this assay (3 µM), but the saturation could not be reached with the used concentrations (Appendix A). The control (1 mM ATP) showed no KRAS G13D stabilization. As the stability increase was saturated in the Protein-Probe assay, the intrinsic nucleotide exchange or another factor affecting nucleotide binding was most likely responsible. The high nucleotide concentration in solution and the high nucleotide binding affinity of KRAS together change the equilibrium and drive RAS to stay constantly in a nucleotide-loaded and thus stable form. As the stabilization is RAS concentration-dependent, with the used nucleotide concentrations, it was detected only with the Protein-Probe, using 50 nM KRAS.

To further test nucleotide-induced KRAS stabilization, we next performed single or dual (10 or 100 µM) concentration stability assays using GDP, GTP (guanosine-5’-triphosphate), and ATP (adenosine-5’-triphosphate). Assays were performed not only with the G13D mutant but also several other KRAS mutants. With KRAS G13D, the addition of 10 µM GDP shifted the observed T_m_ value by over 10 °C, which was more than with KRAS WT or G12D (Figure 2B, Appendix A). More significantly, no change in T_m_ was observed with Q61R. These stabilizing concentrations of GDP are several-fold higher compared to the reported affinity to KRAS, which indicates stabilization potentially in a Mg^2+^-independent fashion. Indeed, GDP stabilized KRAS WT both in the presence and absence of Mg^2+^, and the effect was more pronounced in the presence of 1 mM EDTA, resulting in a ΔT_m_ of 11.3 °C compared with a ΔT_m_ of only 7.6 °C in 1 mM MgCl_2_ (Figure 2C, Appendix A) [40,43,45]. Another divalent cation, Ca^2+^, stabilized KRAS WT both with and without GDP [45,46]. When KRAS WT assayed with 1 mM EDTA (T_m_ 50.8 ± 0.6) was compared to the samples with 0.1 or 1 mM CaCl_2_ or 1 mM MgCl_2_, the stability increase was 1.2, 3.7, and 9.1 °C, respectively. On the other hand, GDP addition brought the T_m_ with both 0.1 and 1 mM Ca^2+^ to the same value, 61.2 ± 0.9 and 61.8 ± 0.6, respectively (Figure 2C). This indicates that Ca^2+^ cannot be used as a replacement for Mg^2+^ but that stability increase is most probably due to an additional weak affinity binding pocket for Ca^2+^ [46,47]. However, the Ca^2+^ binding affinity to this pocket is expected to be biochemically too weak to play a role, especially considering the low Ca^2+^ concentrations in the cytoplasm.

In addition to KRAS, we tested the additional small GTPases NRAS, HRAS, and RhoA, as well as Gαi, an α-subunit from the heterotrimeric G protein, to see if similar behavior in the presence or absence of Mg^2+^ and GDP/GTP occurred. Again, ATP showed no stabilization at any tested condition or protein (data not shown). On the other hand, all the tested small GTPases were stabilized by GDP and GTP both with and without Mg^2+^, as shown with RhoA and the Protein-Probe technique (Figure 2D and Appendix A). RhoA was significantly more stabilized with GTP than GDP, and the preference further increased in the absence of Mg^2+^, as also confirmed with SYPRO Orange and ANS dyes (Appendix A) [48]. In the presence of EDTA, NRAS and potentially also HRAS showed preference for GTP over GDP, indicating higher binding affinity of GTP in the absence of Mg^2+^, but the effect was less pronounced than with RhoA (Appendix A) [40,43,48].

To further confirm that KRAS stability is linked to its intrinsic enzymatic activities and affinity for the nucleotide, we performed an assay with KRAS V14I located in a P-loop [49]. In addition to increased binding to SOS, it also had increased intrinsic nucleotide dissociation properties. As expected, KRAS V14I had a relatively low thermal stability (T_m_ ~48 °C), and it was stabilized with both nucleotides but showed a clear preference for GTP over GDP (Appendix A). On the other hand, Gαi showed no clear stabilization with Mg^2+^ (max ΔT_m_ = 1.0 °C) or without Mg^2+^ (max ΔT_m_ = 3.3 °C) when 100 µM GDP or GTP was tested (Appendix A). In addition, the effect of Mg^2+^ with Gαi was lower (~4.2 °C) compared to, e.g., KRAS assayed in the same conditions.

In biochemical and enzymatic studies with KRAS, non-hydrolysable GTP analogs are often used, as GTP-loaded RAS has a limited shelf-life. These analogs, however, have a lower affinity compared to the natural GTP ligand, and they also affect RAS conformation. We hypothesized that the use of different GTP analogs change the KRAS stability differently, which needs to be considered when these results are interpreted. Initially, we performed a nucleotide titration using a SOS^cat^ catalyzed nucleotide exchange assay with KRAS (Figure 3A). The assay was performed with 15 nM KRAS, which limits us in monitoring the binding affinity of the highest affinity nucleotides, GDP (EC_50_ = 11.9 ± 0.7 nM), GTP (EC_50_ = 7.9 ± 0.6 nM), and GTPγS (guanosine-5’-(γ-thio)-triphosphate) (EC_50_ = 11.7 ± 0.8 nM) (Appendix A). In contrast, GMP-PNP (guanosine-5’-[(β,γ)-imido]triphosphate), GMP-PCP (guanosine-5’-[(β,γ)-methyleno]triphosphate), and especially GMP had significantly lower affinity for KRAS, with EC_50_ values of 71.2 ± 9.3 nM, 542 ± 38 nM, and 101 ± 8 µM, respectively (Figure 3A, Appendix A). Protein-Probe analysis with the same analogs and KRAS WT showed a similar pattern as nucleotide exchange (Figure 3B). The high affinity nucleotides GDP, GTP, and GTPγS stabilized KRAS already at 10 µM concentration, while for GMP-PNP, 300 µM concentration was needed for stabilization. The 300 µM GMP-PCP showed minor effects, but GMP, often used to stabilize the nucleotide-free form of RAS, had no KRAS stabilizing effect even at 1 mM concentration. Because we clearly observed stability increasing effects with GTP analogs in solution, we next loaded KRAS with GMP-PNP, the most widely used GTP analog due to its resistance to hydrolysis. As shown previously, a significantly reduced T_m_ value, 53.2 ± 0.3, was detected with GMPPNP-loaded KRAS in comparison to GDP-KRAS, 60.4 ± 0.2 (Figure 3C) [20]. Thus, we can conclude that not only assay conditions, but also mutations, activation state, and nucleotides/analogs affect the observed T_m_ and must be considered when comparing the results obtained using any TSA method.

### 2.3. Small Molecular Covalent Inhibitors Are Overrepresented in TSA over Non-Covalent Inhibitors

TSA is not only used as a tool for protein stability but also for protein–ligand interaction (PLI) studies utilizing the thermal shift that occurs upon binding. KRAS is a high-priority drug target, but the modulation of its functionality has been difficult [50,51]. This is due to the lack of an obvious targetable binding pocket, except for the nucleotide binding one. However, drugging KRAS G12C has shown promise, as this mutant enables covalent intervention by small molecular inhibitors [50,51]. As multiple parameters related to assay conditions might affect RAS stability, we next studied KRAS with multiple known covalent or non-covalent inhibitors to address their function side by side under the same conditions. 

Selected inhibitors were first assayed with a conventional nucleotide exchange assay utilizing KRAS WT and its G12C mutant [23,25]. This was done to ensure the correct functionality and concentration range for the inhibitors. Two selected covalent binders, ARS-853 and ARS-1620, were used with KRAS G12C, and a non-covalent inhibitor, BI-2852, with KRAS WT. As a control for both assays, we used the KRAS/SOS interaction inhibitor BAY-293 and the non-binding control BI-2853. All inhibitors were tested in a nucleotide exchange assay using 100 nM KRAS (WT or G12C) and 10 nM SOS^cat^. Importantly, the individual inhibitors were preincubated with KRAS for 20 min before assay initiation by addition of SOS^cat^. In the selected conditions, BAY-293 showed KRAS mutation-independent inhibition, with observed IC_50_ values of 11.5 ± 2.2 and 17.5 ± 3.0 nM for WT and G12C, respectively (Appendix A, Appendix A). BI-2852, but not BI-2853, inhibited KRAS WT nucleotide exchange with the observed IC_50_ value of 2.8 ± 0.7 µM. ARS-853 and ARS-1620 inhibited KRAS G12C nucleotide exchange, giving IC_50_ values of 1.1 ± 0.1 and 0.21 ± 0.04 µM, respectively (Appendix A, Appendix A). Based on these results, all inhibitors showed the expected function at the expected concentration range.

Next, thermal stabilization of KRAS G12C by covalent binders was investigated. As expected, both ARS-853 and ARS-1620 showed a significant concentration-dependent thermal stability increase over KRAS G12C alone (Figure 4A). With 50 nM KRAS G12C, saturated stabilization was observed at 5 µM ARS-853 and ARS-1620. At a concentration of 1.25 µM per compound, two thermal transitions could be observed, corresponding to free and covalently compound occupied KRAS G12C (Figure 4A and Appendix A). We could further confirm the specificity of ARS-853 and ARS-1620 towards KRAS G12C over WT when using SYPRO Orange (Appendix A). As expected, the SOS binding BAY-293 and the non-binding BI-2853 gave no response, but also with BI-2852 and KRAS G12C, G12D or WT the thermal stability effect was negligible (Appendix A). ARS-853 and ARS-1620 demonstrated the power of using the more sensitive Protein-Probe technique in comparison to SYPRO Orange, as it enables the estimation of inhibitor binding affinity. Due to the 200-fold lower KRAS G12C concentration in the Protein-Probe assay, 5 µM ARS-853 and ARS-1620 fully saturated the thermal stability increase, giving a ΔT_m_ of 16 to 18 °C, respectively, while with SYPRO Orange and 10 µM KRAS G12C, 20 µM ARS-853 and ARS-1620 still provided double transitions (Figure 4A and Appendix A).

We further continued the study by using two clinical stage KRAS G12C inhibitors, AMG-510 and MRTX849 [50]. In a nucleotide exchange assay with 15 nM KRAS G12C, both AMG-510 and MRTX849 showed very potent binding, giving IC_50_ values of 20.7 ± 1.2 and 6.9 ± 0.8 nM already after ~10 min incubation (Appendix A, Appendix A). Next, we continued by studying KRAS G12C stability with the Protein-Probe (Figure 4B) and with SYPRO Orange and ANS (Figure 4C). With AMG-510, we observed significant signal quenching with ANS or SYPRO Orange, which also was quenched with MRTX849 (Figure 4B). However, this AMG-510-induced signal quenching was non-visible at nM concentrations already producing a maximal thermal stability increase in the Protein-Probe assay (Figure 4C). Even though the affinity for KRAS G12C was higher for MRTX849 than for AMG-510, the thermal stability increase in the Protein-Probe assay with AMG-510 was more pronounced than with MRTX849, giving ΔT_m_ values of 24.8 vs. 15.6 °C, respectively. The confirmed trend was the same with other dyes, and this is likely due to differences in the binding modes of these molecules, as both target the same KRAS pocket [50,51]. The effect on signal quenching seems to be indicative for binding, as the SYPRO Orange fluorescence was nearly preserved with the KRAS WT (Appendix A). As the signal quenching does not occur directly through SYPRO Orange and AMG510, we cannot rule out the possibility that instead of quenching, AMG-510 shifts the thermal curve out from the measurement range. In our hands, the reliable measurement range ends at approximately 90 °C, and AMG-510 had a tendency to increase the signal at the high-end temperatures. In all cases, the improved sensitivity of the Protein-Probe avoids the complications faced with AMG-510 (Figure 4).

### 2.4. Protein Based Inhibitors Can Induce Covalent-Like Thermal Stability Increase

Biologics, typically proteins, are interesting alternatives, especially when the target does not allow the use of small molecule drugs. RAS has been a notoriously difficult target for small molecule drugs due to a lack of defined binding pockets, supporting the use of biologics that provide different modes of action. Thus, we decided to test Designed Ankyrin Repeat Proteins (DARPins). They are highly stable binding molecules selected from diverse synthetic libraries to bind specifically and with high affinity to the target of interest [52]. As they contain no cysteine and can fold in the cytoplasm, they have potential as intracellular inhibitors. For the study, three different DARPins, K27, K13, and K19, were tested to define their protein–protein interaction (PPI) effect on KRAS stability [53,54].

For DARPins K27, K13, and K19, IC_50_ values of 30.6 ± 4.2, 140 ± 23, and 125 ± 16 nM were observed, respectively, in a nucleotide exchange assay (Appendix A, Appendix A). Next we utilized this information to monitor the DARPin-induced increase in thermal stability, which we previously reported with K27 and the Protein-Probe method [20]. To our surprise, K13 and K19 had no effect on KRAS thermal stability in any of the tested concentrations, while K27 showed the expected increase in KRAS thermal stability (Figure 5) [20]. K13 and K19 bind to the same position on KRAS, the allosteric lobe at the interface between helix α3/loop 7/helix α4 [54]. This area is distant from the area where K27 binds, namely the switch I area overlapping the SOS binding site [53]. Due to these differences, we continued to monitor the effect on SOS^cat^-induced and intrinsic nucleotide exchange with these DARPins. All these DARPins blocked the SOS^cat^-induced reaction (Appendix A), but only K27 also blocked the intrinsic nucleotide exchange activity of KRAS (Appendix A). This indicates that K27 potentially has a more significant effect on the KRAS structure in comparison to K13 and K19, acting mainly by blocking the KRAS interaction with SOS^cat^ [53,54]. By using KRAS WT and K27 as a control, we further tested if K27 would stabilize the G12D, G13D, Q61L, and Q61R mutants. Of these mutants, only KRAS Q61R was not stabilized by K27, further supporting a link between intrinsic KRAS nucleotide exchange activity and thermal stability (Figure 5B and Appendix A).

Q61 is part of the Switch II region, which plays an essential role in effector interactions, GAP-mediated and intrinsic GTP hydrolysis, as well as nucleotide binding [38]. Thus, we next tested if the removal of Mg^2+^ has an effect on K27-induced KRAS stabilization. K27 is reported as a GDP-KRAS binder, and without Mg^2+^, we detected no thermal stability increase with K27 and KRAS WT (Figure 5C). Additionally, ARS-1620 also required Mg^2+^ to stabilize KRAS G12C (Appendix A). These results indicate that TSA results with KRAS might be biased toward those binders that cover the Switch I and II and/or that have an effect on nucleotide loading state. This is in line with the structural data showing no effect on these areas when KRAS is bound to K19 [54]. Unfortunately, this statement could not be confirmed by other TSA dyes, due to their lack of sensitivity and concentration restrictions. It is unlikely, however, that this finding would be only related to the Protein-Probe assay, but it needs to be considered prior to assay design, as assays run with or without Mg^2+^ are expected to produce differences in results.

### 2.5. Non-Observable Thermal Shifts Can Be Visualized Using a Competition Thermal Stability Protocol

Thermal stability changes cannot be used to visualize all KRAS binding inhibitors, as found, e.g., for DARPins K13 and K19. The binding area for K13 and K19 is known and their binding can be expected to affect, in turn, the binding of KRAS G12C to covalent inhibitors targeting the switch-II pocket. Therefore, MRTX849, AMG-510, ARS853, and ARS1620 were selected to investigate a competition-based protocol. We hypothesized that upon K13 or K19 binding, the large thermal stability increase induced by these covalent binders will be reduced or disappear, as the DARPin can block the binding of the covalent inhibitor either competitively or by changing the KRAS structure. We first tested this hypothesis by using 1:1 complex of KRAS (50 nM) and K13 (50 nM), and we found this to be true especially with AMG-510 and to lesser extent with MRTX849 (Figure 6). In the presence of K13, 100 nM AMG-510 had no effect on KRAS stability, but it was detected at higher AMG-510 concentrations (Figure 6A). Change from a single thermal transition curve with 900 nM AMG-510 to double transition curves in the presence of K13 is also a clear indication for competitive binding. On the other hand, the ΔT_m_ was only 3–7.4 °C with MRTX849, in comparison to ΔT_m_ values over 10 °C monitored with AMG-510 using the same protocol. These changes observed with either AMG-510 or MRTX849 may indicate slight differences in binding and conformational compatibility with the K19 DARPin-bound KRAS. With ARS853 and ARS1620, we also found the affinity-driven effect, and both inhibitors showed competitive behavior (Appendix A). In summary, by using a competitive protocol, one could potentially help to find compounds that target specific pockets on the target protein. However, affinity restrictions set by the TSA method used and chosen binders must be carefully considered.

## 3. Materials and Methods

### 3.1. Protein-Probe Thermal Stability Assays

Detailed lists of materials and instrumentation as well as the production/purification of KRAS and related proteins are presented in the Appendix A. In addition, detailed protocols for RAS nucleotide exchange assays and data analysis are presented there. All presented assays were performed in triplicate unless otherwise indicated.

All Protein-Probe assays were performed using a sample volume of 8 µL and a two-step assay protocol. Detection was performed at room temperature (RT) by adding 65 µL Protein-Probe solution: 4 µM HIDC (1,1,3,3,3′,3′-hexamethylindodicarbocyanine iodide) and 1–1.5 nM Eu^3+^-labeled peptide probe (NH_2_-EYEEEEEVEEEVEEE) [19]. Assays were performed in a sample buffer containing 10 mM HEPES (pH 7.5), 0.001% Triton X-100 and 20 mM NaCl and the buffer was supplemented with 0–10 mM MgCl_2_, 0–1 mM CaCl_2_, or 0–1 mM EDTA, depending on the assay. In all thermal stability assays, samples were preincubated at RT for 10 min (nucleotides) or 30 min (inhibitors) prior to thermal measurements. Depending on the studied protein, thermal ramping was performed up to 95 °C, using 5 °C intervals. At each temperature, samples were incubated for 3 min before the Protein-Probe solution was added, and the TRL-signals at 620 nm were monitored at RT.

### 3.2. Concentration and Buffer Composition–Related Effects on GTPase Stability

Melting curves of KRAS WT (50–1250 nM), GMPPNP-KRAS (50 nM), and mutants G13D (50–150 nM) and Q61R (50–1250 nM) were monitored in sample buffer with or without 1 mM MgCl_2_ or 1 mM EDTA. In addition, KRAS WT was monitored in a sample buffer supplemented with 0–10 mM MgCl_2_ or 0–1 mM CaCl_2_. The melting curves for HRAS (1000–2000 nM), NRAS (500–2000 nM), Ac-KRAS (100–2000 nM), iMet-KRAS (100–1000 nM), RhoA (2 µM), Gαi (50 nM), and KRAS mutants Q61L (50–1250 nM), G12D (50–150 nM), G12C (10–1250 nM), and Q61R (50–150 nM) were monitored in the presence of 1 mM MgCl_2_.

### 3.3. Nucleotide Concentration–Induced GTPase Stabilization

KRAS G13D (50 nM) was monitored in the presence of 0–900 nM GDP, and KRAS WT, G13D, G12D, and Q61R (50 nM) were monitored with 10 and/or 100 µM GDP, GTP, and ATP in buffer supplemented with 1 mM MgCl_2_. KRAS WT (50 nM) was also monitored with 10 µM GDP in a buffer without MgCl_2_ or supplemented with 0–1 mM CaCl_2_ or 1 mM EDTA. HRAS (2 µM), NRAS (2 µM), RhoA (2 µM), and Gαi (50 nM) were monitored with 100 µM GDP, GTP, or ATP in buffers with or without 1 mM MgCl_2_ and 1 mM EDTA. In addition, melting curves for KRAS WT (50 nM) were measured with GTP analogs GTPγS, GMPPNP, and GMPPCP (0–300 µM), and GMP (0–1000 µM) in the presence of 1 mM MgCl_2_.

### 3.4. KRAS Thermal Stabilization Using Small Molecular Inhibitors

All inhibitor assays were performed with a buffer containing 1 mM MgCl_2_. Thermal curves for KRAS G12C and WT (50 nM) were measured in the presence of ARS-853 (0–5000 nM), ARS-1620 (0–5000 nM), AMG-510 (0–900 nM), and MRTX849 (0–900 nM). KRAS WT, G12C, and G12D (50 nM) were also monitored with 10 µM BI-2852, BI-2853, and BAY-293. In addition, KRAS G12C (50 nM) was studied with ARS-1620 (1 µM) in a buffer without MgCl_2_ and with 1 mM EDTA.

### 3.5. KRAS Thermal Stabilization Using Protein-Based Binders

Assays were performed using three DARPins alone or in combination with small molecular inhibitors. DARPins K13 (200 nM) and K19 (200 or 500 nM) were assayed with KRAS WT and G12C (50 nM) and DARPin K27 (100 nM) KRAS WT, G13D, G12D, Q61R, and Q61L (50 nM) in a buffer supplemented with 1 mM MgCl_2_. KRAS WT (50 nM) was also assayed with DARPin K27 (50 nM) with 0–1 mM EDTA. Competitive thermal stability assays were performed by first combining KRAS G12C (50 nM) with DARPin K13 (50 nM) and after 10 min incubation by additionally adding AMG-510 (0–900 nM), MRTX849 (0–900 nM), ARS-853 (0–20 µM), or ARS-1620 (0–20 µM). The thermal curves were monitored after an additional 30 min incubation at RT.

### 3.6. SYPRO Orange and ANS Thermal Ramping Assays

SYPRO Orange and ANS assays were performed using a one-step protocol, where the samples (8 µL) and SYPRO Orange/ANS solution (12 µL) were combined prior to the first heating step. The samples were incubated for 3 min at each temperature, followed by fluorescence signal measurement. SYPRO Orange was used at 5× (stock 5000×) and ANS at 10 µM final concentration.

Thermal stability curves for KRAS WT, G12C, G13D, and Q61R (0–30 µM) were monitored both with SYPRO Orange and ANS in an assay buffer with or without 1 mM MgCl_2_. In addition, HRAS, NRAS, KRAS V14I, RhoA, and Gαi were studied at 10 µM concentration. KRAS G13D (3 µM) was assayed with SYPRO Orange in thermal ramping in combination with 0–1000 µM GDP, 1000 µM GTP, and 1000 µM ATP in a buffer supplemented with 1 mM MgCl_2_. Additionally, 10 µM HRAS, NRAS, KRAS V14I, Ac-KRAS iMet-KRAS, RhoA, and Gαi were studied without additional nucleotides or with 10 or 100 µM GDP/GTP in an assay buffer supplemented with 1 mM MgCl_2_ or 1 mM EDTA.

KRAS WT, G12D, and G12C (3 µM) were monitored with 20 µM BI-2852, BI-2853, and BAY-293, and 10 µM KRAS WT and G12C were also measured in combination with 20 µM ARS-853 and ARS-1620. In addition, KRAS G12C (10 µM) was assayed with 0–20 µM AMG-510 and MRTX849 using both SYPRO Orange and ANS.

## 4. Conclusions

In this study, we investigated KRAS and additional small GTPases in terms of thermal stability to critically evaluate TSA suitability for inhibitor screening and validation. TSA is a widely used tool for protein stability monitoring, but some critical factors influence the assay. Based on our findings, in addition to buffer composition, the variables of free nucleotide, protein concentration, and inhibitor binding mode/area all affect the TSA data. Direct comparison with traditional nucleotide exchange assays showed that inhibitors blocking either intrinsic nucleotide exchange or binding covalently to the target GTPase showed significant thermal shifts. However, other inhibitors that are functional in a nucleotide exchange assay with SOS^cat^ showed only small or negligible thermal shifts. Nevertheless, we showed that some of these binders can be validated through a competitive TSA protocol, by using the Protein-Probe technique. This further proves that molecules can bind without evident target stabilization. TSA can be a valuable tool both for inhibitor screening and validation, but one must appreciate that not all interactions can be detected. In addition, the results might be biased for inhibitors functioning in a certain way, as we showed with KRAS in a buffer with or without MgCl_2_. It is clear that data obtained with minor changes in buffer composition might be affected. Moreover, this should be taken into account especially in the case of studies using high protein concentrations, in which the carried storage buffer can also affect the results. These factors must all be considered carefully when designing a TSA screen so that their effect can be considered during data interpretation.

## Figures and Tables

**Figure 1 ijms-23-07095-f001:**
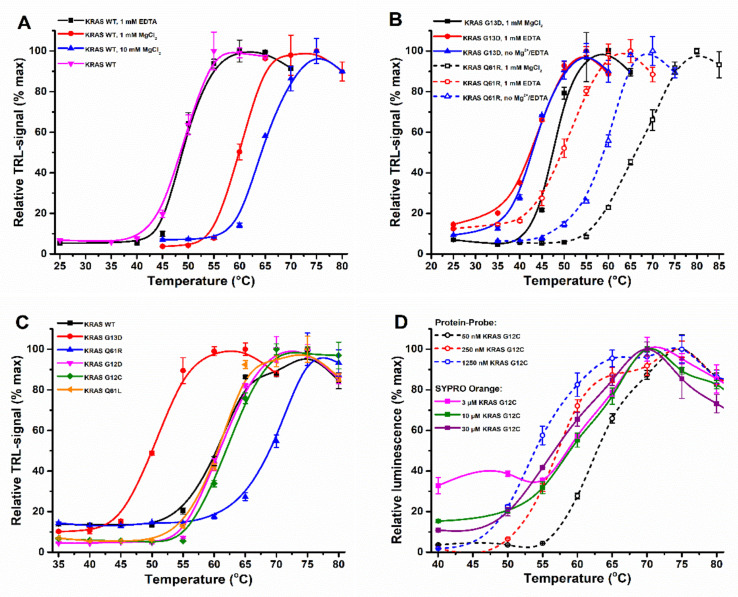
Buffer, mutational status, and concentration effects on KRAS thermal stability. (**A**) KRAS WT (100 nM) showed clear MgCl_2_-dependent thermal stabilization when assayed using the Protein-Probe. The presence of 1 (red) or 10 mM (blue) MgCl_2_ increased KRAS WT thermal stability by approximately 10 °C over that without MgCl_2_ (black) and with additional 1 mM EDTA (magenta). (**B**) KRAS (100 nM) mutational status affects the MgCl_2_ dependence. KRAS G13D (solid line) and Q61R (dashed line) responded differently in the absence of MgCl_2_ (blue) or presence of 1 mM MgCl_2_ (black) or 1 mM EDTA (red). While KRAS G13D is thermally unstable in all conditions, Q61R showed significantly compromised stability in the presence of 1 mM EDTA. (**C**) KRAS (50 nM) thermal stability is dependent on its mutational status. From the hotspot KRAS mutants tested, G13D (red) and Q61R (blue) significantly differ from the other tested KRAS proteins: WT (black), G12D (magenta), G12C (green), and Q61L (orange). (**D**) Increase in KRAS G12C concentration decreased its stability in Protein-Probe (dashed line) and SYPRO Orange (solid line) assays. Using the Protein-Probe, a significant decrease in KRAS G12C thermal stability was obtained when protein concentration was increased from 50 nM (black) to 250 nM (red) or 1250 nM (blue). Using the SYPRO Orange, the decrease in KRAS G12C T_m_ was less pronounced at the assayed concentrations of 3 µM (magenta), 10 µM (green) or 30 µM (violet). Data represent mean ± SD (*n* = 3).

**Figure 2 ijms-23-07095-f002:**
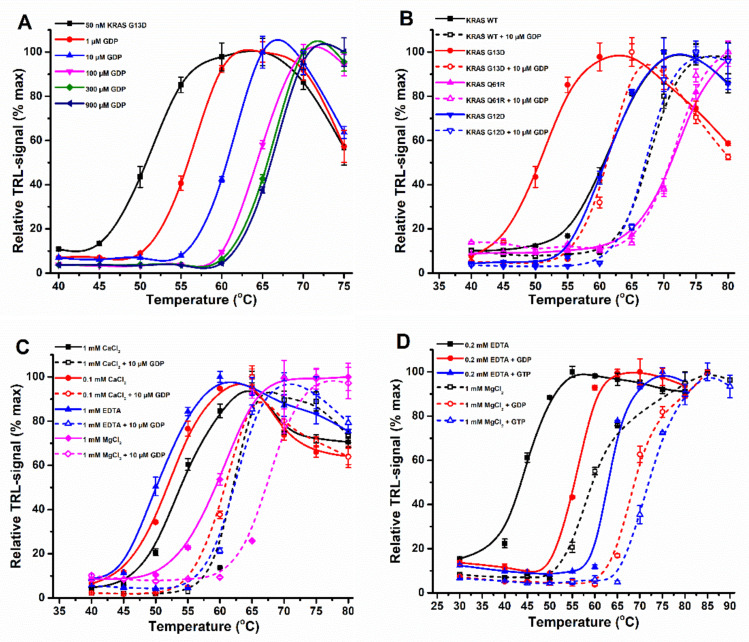
KRAS and other small GTPases were stabilized by GDP/GTP in the Protein-Probe assay. (**A**) KRAS G13D (50 nM) stability was determined in the presence of free GDP (0–900 µM). KRAS G13D (black) was significantly thermally stabilized with 1 µM GDP (red). GDP-induced stabilization of KRAS G13D further increased with 10 (blue) and 100 µM (magenta) GDP and was saturated at approximately 300 µM (green) concentration, without further change with 900 µM GDP (navy). (**B**) KRAS (50 nM) thermal stabilization in the absence (solid line) or presence (dashed line) of additional GDP (10 µM) was dependent on its mutational status. KRAS G13D (red) was more responsive to GDP addition in comparison to KRAS WT (black) or G12D (blue). KRAS Q61R (magenta) was not stabilized by the addition of GDP. (**C**) GDP-induced KRAS WT (50 nM) stabilization was not dependent on divalent cations. Both 0.1 (red) and 1 mM (black) Ca^2+^ stabilized KRAS, but to a lesser extent than 1 mM Mg^2+^ (magenta), when compared to KRAS with 1 mM EDTA (blue). In all buffers, 10 µM GDP (dashed line) addition stabilized KRAS in comparison to buffer without additional nucleotide (solid line). (**D**) RhoA WT (2 µM) nucleotide-specific stabilization was more significant in the presence of 0.2 mM EDTA (solid) than with 1 mM Mg^2+^ (dashed). RhoA (black) stability increased significantly more with 100 µM GTP (blue) than GDP (red). Difference in GTP preferred stability increase was more significant in the absence of Mg^2+^. Data represent mean ± SD (*n* = 3).

**Figure 3 ijms-23-07095-f003:**
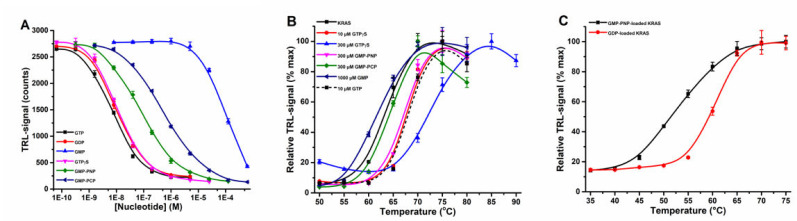
KRAS stabilization by additional nucleotides is affinity dependent. (**A**) The binding affinity order for nucleotide binding to KRAS (15 nM) was determined by QRET nucleotide exchange assay in a SOS^cat^ (10 nM) induced reaction. KRAS binding for GTP (black), GDP (red), and GTPγS (magenta) was <15 nM, while GMP-PNP (green) and GMP-PCP (navy) bound at micromolar and GMP (blue) only at millimolar levels. (**B**) KRAS (50 nM) nucleotide-dependent thermal stabilization was affinity- and concentration-dependent as assayed with the Protein-Probe. KRAS WT (solid black) was similarly stabilized by 10 µM GTP (dashed black) and 10 µM GTPγS (red). At 300 µM concentration, GTPγS (blue) showed the highest stabilization, followed by GMP-PNP (magenta) and GMP-PCP (green). GMP (navy) showed no KRAS stabilization. (**C**) Thermal stability using preloaded KRAS (50 nM) was in agreement with solution-based binding using an excess of nucleotides. KRAS-GDP WT (red) was thermally significantly more stable than KRAS-GMP-PNP WT (black). Data represent mean ± SD (*n* = 3).

**Figure 4 ijms-23-07095-f004:**
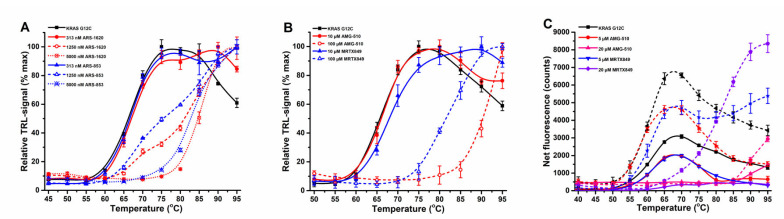
Covalent KRAS G12C inhibitors in the KRAS thermal stability assay. (**A**) 50 nM KRAS G12C (black) was stabilized in a concentration-dependent manner by both ARS-1620 (red) and ARS-853 (blue) inhibitors (313–5000 nM) in the Protein-Probe assay. (**B**) 50 nM KRAS G12C (black) was stabilized in a concentration-dependent manner by AMG-510 (red) and MRTX849 (blue) inhibitors in the Protein-Probe assay. (**C**) 6 µM KRAS G12C (black) was concentration-dependently stabilized by AMG-510 (red scale) and MRTX849 (blue scale) inhibitors in the SYPRO Orange (solid line) and ANS (dashed line) assays. No thermal shift was detected with 5 µM AMG-510, but the second transition was visible with the same concentration of MRTX849. An increase in AMG-510 concentration induced a significant signal quenching of both SYPRO Orange and ANS dyes. Data represent mean ± SD (*n* = 3).

**Figure 5 ijms-23-07095-f005:**
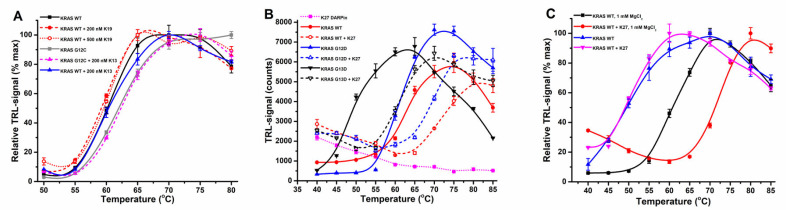
KRAS thermal stabilization by DARPins in the Protein-Probe assay. (**A**) Neither DARPin K13 nor K19 stabilized KRAS WT or G12C (50 nM). K19 (red) did not stabilize KRAS WT (black) at 200 (red dashed) or 500 nM (red dotted). Similarly 200 nM K13 had no effect on KRAS WT (black compared to blue solid lines) or KRAS G12C (gray solid line compared to magenta dashed line). (**B**) DARPin K27 (100 nM) showed a similar degree of stabilization with 50 nM KRAS WT (red), G12D (blue), and G13D (black). DARPin K27 (magenta) was responsible for the signal in the Protein-Probe assay at low temperatures. (**C**) DARPin K27 (100 nM)-induced thermal stabilization of KRAS WT (50 nM) was Mg^2+^-dependent. In addition, 1 mM MgCl_2_ stabilized KRAS (black compared to blue) and was also necessary for K27-induced stabilization (red compared to magenta). Data represent mean ± SD (*n* = 3).

**Figure 6 ijms-23-07095-f006:**
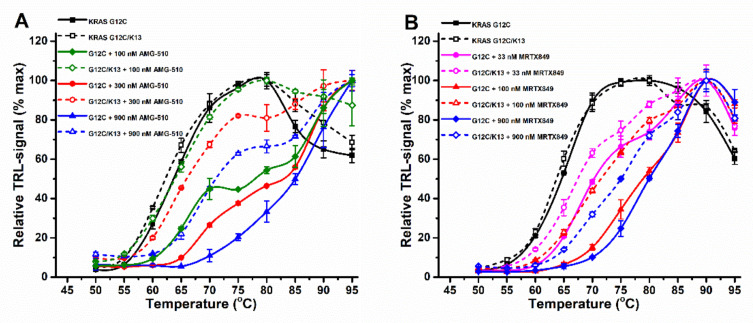
Competitive thermal shift assay for visualizing inhibitors having no direct effect on KRAS thermal stability. (**A**) In the absence of K13 (solid), 50 nM KRAS G12C (black) was significantly stabilized by 100 (green), 300 (red), or 900 nM (blue) AMG-510. In the presence of K13 (dashed), KRAS G12C was not affected, but the stabilizing effect of AMG-510 could be totally or partially reversed. (**B**) In the absence of K13 (solid), 50 nM KRAS G12C (black) was significantly stabilized by 33 (magenta), 100 (red), or 900 nM (blue) MRTX849. In the presence of K13 (dashed), KRAS G12C was not affected, but the stabilizing effect of MRTX849 could be partially reversed. Data represent mean ± SD (*n* = 3).

## Data Availability

Data are available on request to the corresponding author.

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
