# Peer review of "Thermal Shift Assay for Small GTPase Stability Screening: Evaluation and Suitability"

_ijms, 2022, doi:10.3390/ijms23137095_

Round 1

Reviewer 1 Report

  The authors reported that TSA assay with protein-probe is better than other dues because it can work at a lower concentration. In this article, the authors evaluated the assay condition carefully and summarized the result. The results are informative for readers who plan to d a similar assay or use data of TSA for other experiments.

   I wrote some major and minor comments below. 

Major comments

(1)  The authors used many KRAS mutants. The mutants change their property. Presumably, readers could not imagine the relationship between the mutation position and the effect of the protein. The figure of the molecular structure of KRAS and the mapping of mutation would help readers to know the experimental concept and understand the results.

Please add the figure in the supplementary.

(2) The authors showed the many results of TSA and other assays in graphs but wrote the values calculated using the data plots (Tm or EC50 ) in the sentence only. It is too difficult to read. The authors should make a table summarizing the data.

Minor comments

(3) The representation of graphs is not consistent.

   The font size indicating the condition of each plot in Figure 1B is different from the size of other figures in Fgure1. 

Also, the mark (ex.box, triangle) is not the same description  between Figure 1A1&B and 1C1&D. In Figure 1A1B, they are represented without line, but with line in FIgure 1C&D. 

 Two persons might draw the figures together. Please check carefully all figure again and correct them. 

(4) line 95

 at codons -> amino acid?

(5) line 409

 52 -> [52]? 

Author Response

The authors reported that TSA assay with protein-probe is better than other dues because it can work at a lower concentration. In this article, the authors evaluated the assay condition carefully and summarized the result. The results are informative for readers who plan to d a similar assay or use data of TSA for other experiments.

I wrote some major and minor comments below. 

Major comments

(1)  The authors used many KRAS mutants. The mutants change their property. Presumably, readers could not imagine the relationship between the mutation position and the effect of the protein. The figure of the molecular structure of KRAS and the mapping of mutation would help readers to know the experimental concept and understand the results.

Please add the figure in the supplementary.

Thank you for the comment. We have now added the figure showing KRAS structure as Figure S1 and marked the positions of those mutants studied here (M1, G12, G13, V14, and Q61). The structural network is difficult as amino acids affect their neighboring residues, but from the figure it becomes clear that hot spot mutations are close to magnesium and nucleotide binding pocket.

(2) The authors showed the many results of TSA and other assays in graphs but wrote the values calculated using the data plots (Tm or EC50 ) in the sentence only. It is too difficult to read. The authors should make a table summarizing the data.

We agree that values might be difficult to obtain from the text and in some cases the message might be something else as the thermal shift (ΔTm). Also as we have said, the exact Tm values for example might not be highly comparable except within the exactly same assay. In any case, summarizing tables are always valuable when conducted with care, and thus we have now added Tables S1 and S2. In the Table S1 we show the Tm values using the Protein-Probe and two concentrations of the studied protein. We have also indicated the effect of buffer components and nucleotides in the same table. Results are selected with care and the results in 50 nM group and 1 µM group are comparable inside the one group. Some results are not added as there has been minor changes in the protocol or most often protein concentration. Table S2 is more simple as in most cases RAS and SOS concentrations have no effect on the inhibitor/binder behavior. However, for example nucleotide EC50 values cannot be ever monitored with this type of setup in a reliable way, as the sensitivity of the assay should be rather in low pM than low nM level in case of small GTPases. This has been also indicated next to the Table S2.

Minor comments

(3) The representation of graphs is not consistent.

The font size indicating the condition of each plot in Figure 1B is different from the size of other figures in Fgure1. 

Also, the mark (ex.box, triangle) is not the same description  between Figure 1A1&B and 1C1&D. In Figure 1A1B, they are represented without line, but with line in FIgure 1C&D. 

Two persons might draw the figures together. Please check carefully all figure again and correct them. 

We agree that graphs are not consistent. In some cases this has been selected as the space has been limited etc. We have now made changes to improve the consistency and the presentation.

(4) line 95

 at codons -> amino acid?

(5) line 409

 52 -> [52]? 

Corrected accordingly.

Reviewer 2 Report

In the manuscript of Kobra and colleagues protein-probe thermal stability assays have been used to evaluate the stability of GTPases.

Interestingly, several factors that can influence thermal stability of the proteins have been studied, KRAS and additional GTPases were studied and TSA tool has been used and evaluated for the analysis.

I consider the manuscript suitable for publication.

Author Response

We thank you for the kind comment, I hope that the new version has made the manuscript even better.